



# Seasonal transition dates can reveal biases in Arctic sea ice simulations

Abigail Smith[1], Alexandra Jahn[1], and Muyin Wang[2,3]

[1]Department of Atmospheric and Oceanic Sciences and Institute of Arctic and Alpine Research, University of Colorado Boulder
[2]Joint Institute for the Study of the Atmosphere and Ocean, University of Washington
[3]Pacific Marine Environmental Laboratory, National Oceanic and Atmospheric Administration

**Correspondence:** Abigail Smith (abigail.l.smith@colorado.edu)

**Abstract.** Arctic sea ice experiences a dramatic annual cycle, and seasonal ice loss and growth can be characterized by various metrics: melt onset, break-up, opening, freeze onset, freeze-up and closing. By evaluating a range of seasonal sea ice metrics, CMIP6 sea ice simulations can be evaluated in more detail than by using traditional metrics alone, such as sea ice area. We show that models capture the observed asymmetry in seasonal sea ice transitions, with spring ice loss taking about 1.5–2
months longer than fall ice growth. The largest impacts of internal variability are seen in the inflow regions of melt and freeze onset dates, but all metrics show pan-Arctic model spreads exceeding the internal variability. Through climate model evaluation in the context of both observations and internal variability, we show that biases in seasonal transition dates can compensate for other unrealistic aspects of simulated sea ice. In some models, this leads to September sea ice areas in agreement with observations for the wrong reasons.

## 1   Introduction

Metrics of seasonality have been under-utilized in evaluating sea ice in climate models, due to a lack of long-term observational products, the required daily model output and the complexities in defining seasonal Arctic sea ice transitions. However, new process-based metrics for model evaluation are much needed—the spread between climate model projections of sea ice has been on the order of millions of square kilometers in Coupled Model Intercomparison Project (CMIP) Phases 3, 5, and 6
(Stroeve et al., 2012; SIMIP, submitted 2019), while the causes of the model spread remain largely unknown. Furthermore, the sources of model biases can be obscured by models that show realistic sea ice areas for the wrong reasons. Seasonal sea ice transitions can provide additional process-based metrics to assess climate models. Newly available observational data (Steele et al., 2019) and model output from CMIP6 models (Notz et al., 2016) allow such model assessment for the first time. In this study, we assess how different metrics of seasonal sea ice transitions are represented in models and observations, and
evaluate how these metrics can inform our understanding of simulated Arctic sea ice throughout the year. To do this, we utilize observations and sixteen global climate models, including three sets of ensembles with at least 30 members. Using this rich data set, we evaluate model biases in the context of both the observed sea ice state and multiple simulated representations of internal variability.





## 2 Background: Seasonal transitions in the Arctic sea ice cover

Arctic sea ice exhibits a large annual cycle, with a difference of approximately 8 million square kilometers between the maximum area reached in March and the minimum area in September. From spring to fall, the sea ice experiences various stages of transition forced by both the atmosphere and the ocean (Steele et al., 2010; Persson, 2012; Ballinger et al., 2019). In the spring, clouds formed by northward warm air advection trap downwelling longwave radiation, initiating melt on the surface of the sea ice or in the snowpack on top of it (Persson, 2012; Ballinger et al., 2019). As liquid water collects on the

snow and sea ice, it forms melt ponds. Melt ponds increase the albedo of the surface: snow-covered ice has an albedo of 0.85, while the albedo of melt ponds ranges between 0.1-0.5 (Perovich et al., 2002). Shortwave absorption causes thermodynamic ice loss, and regional studies show that top melt dominates during the early summer (Steele et al., 2010). As the ice breaks up, larger areas of open ocean facilitate greater solar absorption (the albedo of open water is 0.07 (Pegau and Paulson, 2001)) and ice divergence. Energy is absorbed by the surface ocean (Timmermans, 2015) and as solar heating declines in the late summer,

ice melt becomes dominated by bottom melt (Steele et al., 2010). After the annual sea ice minimum in September, ice growth begins. Congelation ice growth along existing ice generally begins before frazil ice growth in the open ocean, meaning that areas where ice is retained throughout the summer experience earlier ice growth than areas of open water (Smith and Jahn, 2019). As fall progresses, the Arctic loses shortwave input. Temperatures decline and ice growth continues through the winter, reaching the maximum area in March.

One metric alone cannot capture the range of seasonal transitions seen in the Arctic, so individual transitions have been characterized by many different definitions in both satellite data and models. Seasonal transition metrics are often referred to interchangeably when they are in fact defined in very different ways. Pan-Arctic satellite retrievals of seasonal sea ice transitions are largely based on passive microwave brightness temperatures. Retrieval algorithms have been created to derive pan-Arctic seasonal sea ice metrics, such as melt onset and freeze onset, directly from brightness temperatures for the entire

satellite era (Markus et al., 2009; Drobot and Anderson, 2001; Belchanksy et al., 2004; Bliss and Anderson, 2014; Bliss et al., 2017). Despite ideal spatial and temporal coverage, melt and freeze onset dates are difficult to utilize for model evaluation. This is in part due to the variations between retrieval algorithms, which can introduce large differences in both magnitude and trends of observed melt onset dates (Bliss et al., 2017). Furthermore, brightness temperatures are not simulated in climate models, so model definitions of melt and freeze onset must be based on other simulated variables. There are multiple valid variables

for diagnosing melt and freeze onset, such as surface temperature, thermodynamic ice growth and snowmelt, and the choice of variable has been shown to impact which processes are captured by the dates, as well as their comparability to satellite data (Smith and Jahn, 2019).

Another strategy for defining seasonal sea ice transitions is to create metrics based on ice concentration, a variable that has equally good spatial and temporal satellite data coverage, since satellite-observed ice concentration is derived from passive

microwave brightness temperatures (Comiso et al., 1997). While this introduces some error through sea ice concentration retrieval algorithms (Ivanova et al., 2015), seasonal sea ice metrics based on ice concentration provide more direct comparisons between models and observations than the current comparisons made between melt and freeze onset. Ice break-up, retreat,





freeze-up and advance have been defined using ice concentration data in satellite data (Stammerjohn et al., 2012; Serreze et al., 2016; Stroeve et al., 2016; Bliss et al., 2019) and in model studies (Barnhart et al., 2016; Wang et al., 2018). However, these

studies are often difficult to compare directly, since the definitions themselves vary substantially in terms of the region and date range studied, the selected threshold of ice concentration and the criteria that the threshold must meet (e.g. last day greater than 15% vs. less than 15% two days in a row). In some cases, definitions are also created to fill specific user needs, such as seasonal navigation (Johnson and Eicken, 2016). A selection of previously used metrics defined using ice concentration are described by Table S1 in the Supplement.

## 3   Data and Methods

In this study we use satellite data to evaluate the performance of fifteen CMIP6 models and the Community Earth System Model Large Ensemble (CESM LE) in terms of their seasonal sea ice transitions in the Arctic from 1979–2014. By utilizing model ensembles, we are able to account for the role of internal variability in modeling the seasonality of Arctic sea ice. As there is no single metric that fully describes seasonal sea ice changes, we utilize a variety of metrics that have been developed

for both models and observations. All spatial medians and means of sea ice variables are calculated between 66-84.5°N in both models and satellite data in order to exclude the largest polar hole in satellite data.

### 3.1   Global coupled climate models

CMIP establishes a set of common experiments for global climate model simulations to quantify how the Earth system responds to forcing, as well identify the sources and consequences of model biases (Eyring et al., 2016). This study uses models from

the most recent phase, CMIP6, in order to evaluate the current state of sea ice simulation. Models are selected for analysis based on the availability of two daily sea ice variables: sea ice concentration (CMIP6 variable name: siconc) and the surface temperature of the sea ice or snow on sea ice (CMIP6 variable name: sitemptop). Our study utilizes all CMIP6 models that met this criteria by March 4th, 2020, which includes fifteen models from nine different institutions (ACCESS-CM2, BCC-CSM2-MR, BCC-ESM1, CanESM5, CESM2, CESM2-FV2, CESM2-WACCM, CESM2-WACCM-FV2, CNRM-ESM2-1, CNRM-

CM6-1, EC-Earth3, IPSL-CM6A-LR, MRI-ESM2-0, NorESM2-LM, NorESM2-MM) (Table S2). As the scope of the study is limited to the satellite era, we use the historical forcing experiment from each model for the period that overlaps with satellite data (1979-2014). All models are kept on their native grids to minimize errors related to interpolation and regridding. Ocean and ice model component details for all models are provided in Supplementary Table S2.

 The number of available ensemble members varies by model, with some models providing as few as three members and

others as many as thirty-five with the required daily variables. Here we use the first ensemble member (r1i1p1f1 or r1i1p1f2) from each model for inter-model comparisons and evaluation against satellite data. To assess the internal variability of the seasonal sea ice metrics, the two CMIP6 Models with at least 30 members (CanESM5 and IPSL-CM6A-LR, hereafter referred to as IPSL) are utilized, in addition to the CESM LE. All of the coupled global models have a nominal ocean resolution of 1°. Relevant variables are available at a daily temporal resolution for 40 members in the CESM LE, 35 members in the CanESM5




and 30 members in the IPSL. When evaluating internal variability, we utilize the first 30 members from the CESM LE and CanESM5 for comparison to each other and IPSL in order to standardize the sample size. The results are insensitive to the subsetting of ensemble members (select figures using all available members are provided in the Supplement).

In previous work, the CESM LE was employed to compare multiple model definitions of melt and freeze onset (Smith and Jahn, 2019). Hence, the CESM LE is utilized here to leverage what is already known about modeled seasonal sea ice transitions in evaluating CMIP6 models, even though the CESM1.1 used for the CESM LE is not a CMIP6 model and does not use CMIP6 forcing. Nonetheless, the CESM LE can be compared with the CMIP6 models over the period 1979-2014, as the CMIP5 RCP8.5 forcing is not substantially different from the CMIP6 historical forcing over the period 2006-2014 (O'Neill et al., 2016). Furthermore, the CESM LE is also a useful addition to the CMIP6 models because it adds diversity to the sea ice models used for evaluating internal variability: the CESM LE uses the CICE Version 4.0 sea ice model, while CanESM5 and IPSL both use the Louvain-la-Neuve Sea Ice Model Version (LIM Version 2.0 and LIM Version 3.0 respectively).

## 3.2 Satellite data

In order to evaluate the climate models against observations, we use the Arctic Sea Ice Seasonal Change and Melt/Freeze Climate Indicators from Satellite Data, Version 1 (Steele et al., 2019). This dataset includes seasonal sea ice indicators from March 1st, 1979 through February 27, 2017, derived from sea ice concentration data from the NOAA/NSIDC Climate Data Record of Passive Microwave Sea Ice Concentration and brightness temperature observations from the DMSP SSM/I-SSMIS Daily Polar Gridded Brightness Temperatures. Indicators (referred to here as seasonal sea ice transition metrics) are described in Sect. 3.3. Data are gridded to a 25 km resolution grid. We calculate the sea ice area from the NOAA/NSIDC Climate Data Record of Passive Microwave Sea Ice Concentration, accessed through the Walsh et al. (2019) dataset.

## 3.3 Defining seasonal sea ice transitions

Establishing a set of metrics for studying the seasonality of Arctic sea ice is important for comparing models and observations, as well as interpreting the relationships between transition times and other sea ice characteristics. Here we utilize a range of seasonal sea ice transition dates and periods to study multiple thermodynamic phases of the ice that may be relevant to our physical understanding of the sea ice. These metrics are summarized in Tables 1 and 2.

### 3.3.1 Melt onset, freeze onset and the melt season

The melt season length is defined as the number of days between melt onset and freeze onset. The melt season length has been utilized as a parameter to investigate energy absorption of the Arctic surface ocean and relationships have been found between the melt season length and sea ice extent (Stroeve et al., 2014). The metrics of melt onset and freeze onset are used to describe the first date of continuous sea ice melt and freeze at each grid cell for each year. Melt and freeze onset are meant to capture a change of phase between water and ice. For melt onset, this means water on the surface of the ice or snowpack. For freeze onset, the change of phase refers to either congelation or frazil ice growth.





| Dates | Date range | Timing | Variable | Threshold |
|---|---|---|---|---|
| Melt onset | 1 Jan to 31 December | for 3 days | surface temperature | above -1°C |
| Freeze onset | 29 June to 15 May | for 21 days | surface temperature | below -1.8°C |
| Break-up | 1 March to SIC minimum date | last day | ice concentration | below 15% |
| Freeze-up | SIC minimum date to 28 February | first day | ice concentration | above 15% |
| Opening | 1 March to SIC minimum date | last day | ice concentration | below 80% |
| Closing | SIC minimum date to 28 February | first day | ice concentration | above 80% |

**Table 1.** Definitions of the seasonal transition dates, including the date range, timing criteria, variable and threshold used. Definitions based on ice concentration are designed to be comparable to Steele et al. (2019)

.

| Intra-seasonal periods | # of days between |
|---|---|
| Melt period | Melt onset and opening |
| Freeze period | Freeze onset and freeze-up |
| Seasonal loss-of-ice period | Opening and break-up |
| Seasonal gain-of-ice period | Freeze-up and closing |
| **Inter-seasonal periods** | **# of days between** |
| Melt season | Melt onset and freeze onset |
| Open water period | Break-up and freeze-up |
| Outer ice-free period | Opening and closing |

**Table 2.** Definitions of the periods of time between the seasonal transition dates, including shorter, intra-seasonal periods of transition as well as longer, inter-seasonal periods. The outer ice-free period and the seasonal loss-of-ice and gain-of-ice periods were defined in Steele et al. (2019).

In satellite retrievals, continuous melt and freeze onset are defined using the brightness temperature of the surface because brightness temperature is sensitive to the phase of water (Markus et al., 2009; Steele et al., 2019). Brightness temperatures are collected at the 19V and 37V polarizations from SMMR,SSM/I and SSMIS sensors. Melt and freeze onset dates are derived from weighted brightness temperature parameters to determine early melt and freeze onset and continuous melt and freeze

onset, and the retrieval algorithm (known as PMW) is described fully in Markus et al. (2009). In this study we use continuous melt and freeze onset because these dates are more representative of a seasonal transition in the sea ice compared to early melt onset. The AHRA dataset provides an alternative set of melt onset dates that are derived from passive microwave brightness temperatures using the AHRA retrieval algorithm instead of the PMW retrieval algorithm. However, the AHRA melt onset dates are more representative of early melt (Drobot and Anderson, 2001), so they are not utilized in this study.

Because climate models do not simulate brightness temperatures, another definition must be used to identify continuous melt and freeze onset dates within models. Although there is no single model definition that fully captures the processes represented





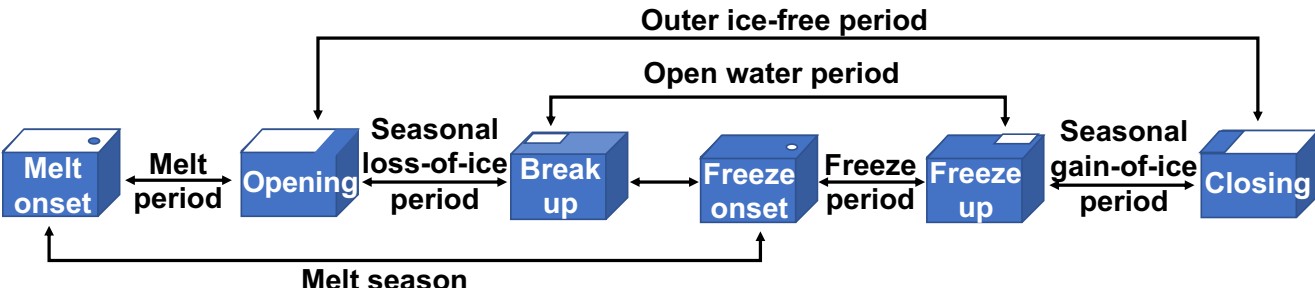

**Figure 1.** Conceptual diagram of seasonal sea ice transitions, beginning with spring melt onset and ending with fall ice closing. Transition dates (Table 1) as well as inter-seasonal and intra-seasonal transition periods (Table 2) are labeled.

by the brightness temperature-based satellite data, recent work demonstrates that melt and freeze onset dates derived from surface temperature are comparable, particularly when considering the range of internal variability (Smith and Jahn, 2019). We therefore utilize model definitions of melt and freeze onset developed in Smith and Jahn (2019), based on the surface

temperature passing below/above a given threshold. For melt onset, a threshold of -1 °C is used to minimize the impacts of daily variability and maintain comparability with previous studies (Jahn et al., 2012; Mortin and Graversen, 2014). For freeze onset, a threshold equal to the freezing point of ocean water (-1.8 °C) is used.

### 3.3.2 Break-up, freeze-up and the open water period

The open water period, also known as the inner ice-free period (Bliss et al., 2019), is defined as the number of days between

ice break-up and freeze-up (also commonly referred to as ice retreat and advance). The open water period has been utilized as a metric to study variability and trends in the sea ice (Serreze et al., 2016; Barnhart et al., 2016) and seasonal predictability of the ice (Stroeve et al., 2016).

    Of the seasonal sea ice transition dates investigated here, the definitions of break-up, freeze-up and the open water period vary the most across the literature (Table S1). In the models, we use the definitions for break-up, freeze-up and the open water

period used in Steele et al. (2019) to allow for comparison with observations. Of the definitions identified and described in Supplementary Table S1, the Steele et al. (2019) definitions are most similar to those established by Stroeve et al. (2016). Break-up is defined as the last day that sea ice concentration passes below the threshold of 15% between March 1 and the annual sea ice concentration minimum date (Bliss et al., 2019). Freeze-up is defined as the first day that sea ice concentration passes above the 15% threshold between the sea ice concentration minimum date and February 28 of the following year. The

open water period is defined as the number of days between break-up and freeze-up.

### 3.3.3 Date of opening, date of closing and the outer ice-free period

The outer ice-free period has been used the least frequently as a metric of Arctic sea ice seasonality, and it is based on the dates of opening and closing defined by Steele et al. (2015). The Steele et al. (2015) definitions are applied in the Steele et al.





(2019) dataset (Bliss et al., 2019). The date of opening is defined as the first day that sea ice concentration passes below the
threshold of 80% between March 1 and the annual sea ice concentration minimum date. Likewise, the date of closing is defined
as the first day that sea ice concentration passes above the 80% threshold between the sea ice concentration minimum date
and February 28 of the following year. By definition, opening must occur before break-up and freeze-up must occur before
closing. However, dates of opening and closing are not limited solely by the existence of break-up and freeze-up dates: if ice
concentration falls below/above 80% but not below/above 15%, there will still be an opening/closing date. This means that the
areal coverage of opening/closing dates is generally larger than those of break-up/freeze-up dates.

### 3.3.4 Melt period and freeze period

In addition to the inter-seasonal periods (melt season, open water period and outer ice-free period) we describe four intra-
seasonal periods between the established dates (melt period, seasonal loss-of-ice period, freeze period and seasonal gain-of-ice
period) (Fig. 1). The melt period is designed to capture the rate of transition between snow and sea ice to the initial appearance
of open water, and it is defined as the number of days between sea ice melt onset and the date of opening. Similarly, the freeze
period is defined as the number of days between freeze onset and freeze-up, and is designed to describe the rate of transition
between initial ice growth water and when an area stops being "ice-free".

### 3.3.5 Seasonal loss-of-ice period and seasonal gain-of-ice period

The seasonal loss-of-ice period and the seasonal gain-of-ice period were established in Steele et al. (2019); Bliss et al. (2019).
The seasonal loss-of-ice period is defined as the number of days between sea ice opening and break-up, and the seasonal gain-
of-ice period is defined as the number of days between freeze-up and closing (Bliss et al., 2019). The seasonal loss-of-ice period
and the seasonal gain-of-ice period can only be calculated at grid cells where both of their respective dates exist for that year
(e.g. both a date of opening and break-up are needed for a valid seasonal loss-of-ice period). The seasonal loss-of-ice period
describes how quickly the ice concentration transitions from 80% to 15%, while the seasonal gain-of-ice period describes the
rate of transition between 15% and 80% ice concentration.

### 3.3.6 Accounting for differences in spatial coverage

Variability between models depends on the selected metric for evaluating seasonal sea ice changes. Over the satellite era,
opening, break-up, freeze-up and closing each have an ice concentration boundary, where there are no existing dates beyond
that boundary, because the ice concentration does not pass the chosen threshold. The models have different sea ice areas
(Supplementary Fig. S1) and the position of the ice concentration boundary varies substantially between them. It is therefore
important to compare the models between each other and the satellite data in a way that captures these differences in spatial
coverage. Using one ensemble member from each CMIP6 model and one ensemble member from the CESM LE, we find the
mean of each characteristic at each grid cell over 1979–2014. We then find the spatial distribution of each grid cell value versus
the fractional area that the value takes up north of 66–84.5°N. This allows for a representation of the pan-Arctic nature of each





185 characteristic without taking a pan-Arctic mean, which would obscure many spatial differences. To quantitatively compare the models to each other and to observations in a pan-Arctic sense, we take the median of the resulting distribution, shown as a histogram in Fig. 2. This value is referred to as the "satellite-era median". Figures 3–8 show each of the seasonal sea ice metrics derived from satellite data, one ensemble member from each available CMIP6 model and one ensemble member of the CESM LE averaged over 1979–2014. Each figure includes stippling to show where the characteristic exists for less than 20% of years

190 in the time period.

**Figure 2.** Area distributions of the average of each metric from 1979–2014: (a) melt onset (b) opening (c) break-up (d) freeze onset (e) freeze-up and (f) closing. Metrics are averaged from 66-84.5°N for satellite data (filled gray) and the first ensemble member of each model (all other colors). All models and satellite data are represented in each panel (a)-(f), but the color labels are distributed across panels (a)-(c).

## 4   Results

Results are presented in five sections. In Sect. 4.1–4.3 we describe the pan-Arctic observed and simulated seasonal sea ice transition metrics from 1979-2014. In Sect. 4.4 and 4.5 we compare observed and simulated relationships between the various seasonal sea ice transition metrics and sea ice area and thickness.

**Figure 3.** Melt onset dates averaged over 1979–2014 at each grid cell using satellite data (a), the first ensemble member of the CESM LE (b) and the first ensemble member of each CMIP6 model (c-q). Stippling indicates where melt onset dates exist in less than 20% of years in the time range. Models on tripolar grids produce plot gaps filled by gray lines.

## 4.1   Spring transitions

We find that the transition from sea ice melt onset to break-up takes two to three months in both satellite data and models. Satellite data show that melt onset generally occurs between April and June over most regions of the Arctic (Fig. 3a), with the



median date of melt onset occurring on May 30. The median date of opening (July 8) occurs about 40 days after melt onset and the median break-up date (July 28) occurs 20 days after opening (Table 3). This indicates that the most time-consuming aspect of the observed spring ice loss is the transition between water formation on the ice or snow surface and a decline in ice area (the melt period). Once open water is present in the grid cell, the transition between 80% ice concentration and 15% ice concentration is faster, due to energy absorption from the change in the surface albedo (Perovich et al., 2002).

**Figure 4.** Opening dates (80% ice concentration threshold) averaged over 1979–2014 at each grid cell using satellite data (a), the first ensemble member of the CESM LE (b) and the first ensemble member of each CMIP6 model (c-q). Stippling indicates where opening dates exist in less than 20% of years in the time range. Models on tripolar grids produce plot gaps filled by gray lines.

Models generally agree with satellite data on the timing of spring transitions (Figs. 3–5), with median melt onset dates over the satellite era occurring between May 15–June 17 (observed median date of May 30) (Table 3). Excluding the CNRM models (which show particularly late median melt onset dates and are explored further in Sect. 4.5), the model spread (May 15–June



**Figure 5.** Break-up dates (15% ice concentration threshold) averaged over 1979–2014 at each grid cell using satellite data (a), the first ensemble member of the CESM LE (b) and the first ensemble member of each CMIP6 model (c-q). Stippling indicates where break-up dates exist in less than 20% of years in the time range. Models on tripolar grids produce plot gaps filled by gray lines.

3) shifts earlier toward the satellite data. Melt onset dates in the inflow regions that fall between January-March demonstrate that melt onset at the surface of the snow pack can occur while the ice area is still expanding in those regions. This highlights that surface temperature based definitions, such as melt onset, capture different physical processes than sea ice concentration-based definitions, as was previously shown (Smith and Jahn, 2019). In agreement with observations, all models project the median length of the melt period to be longer than the seasonal loss-of-ice period. The median time between melt onset and opening (the melt period) is 28–53 days in models and 31 days in the satellite data, while the median time between opening and break-up (the seasonal loss-of-ice period) is 13–33 days in models and 20 days in observations.



|  | Melt onset | Opening (80%) | Break-up (15%) | Freeze onset | Freeze-up (15%) | Closing (80%) |
|---|---|---|---|---|---|---|
| ACCESS-CM2 | Jun 2 | Jul 14 | Jul 31 | Oct 3 | Oct 15 | Oct 11 |
| BCC-CSM2-MR | May 24 | Jul 15 | Jul 27 | Oct 7 | Oct 11 | Oct 10 |
| BCC-ESM1 | May 29 | Jul 22 | Jul 31 | Oct 1 | Oct 10 | Oct 7 |
| CanESM5 | Jun 3 | Jul 11 | Jul 21 | Oct 14 | Oct 16 | Oct 15 |
| CESM2 | May 20 | Jul 7 | Jul 21 | Oct 23 | Oct 28 | Oct 29 |
| CESM2-FV2 | May 20 | Jul 12 | Jul 25 | Oct 16 | Oct 20 | Oct 16 |
| CESM2-WACCM | May 22 | Jul 15 | Aug 3 | Oct 16 | Oct 25 | Oct 22 |
| CESM2-WACCM-FV2 | May 29 | Jul 19 | Jul 28 | Oct 7 | Oct 13 | Oct 6 |
| CNRM-ESM2-1 | Jun 13 | Jul 18 | Jul 27 | Oct 25 | Oct 29 | Nov 5 |
| CNRM-CM6-1 | Jun 17 | Jul 18 | Jul 28 | Oct 18 | Oct 24 | Oct 29 |
| EC-Earth3 | Jun 1 | Jul 9 | Jul 21 | Oct 10 | Oct 8 | Oct 4 |
| IPSL-CM6A-LR | May 15 | Jul 5 | Jul 22 | Nov 2 | Oct 23 | Oct 25 |
| MRI-ESM2-0 | May 22 | Jul 6 | Jul 28 | Oct 23 | Oct 24 | Oct 25 |
| NorESM2-LM | May 21 | Jul 14 | Aug 3 | Oct 15 | Oct 23 | Oct 22 |
| NorESM2-MM | May 20 | Jul 14 | Jul 31 | Oct 16 | Oct 25 | Oct 22 |
| CESM LE | May 21 | Jul 15 | Jul 28 | Sep 28 | Oct 14 | Oct 5 |
|  |  |  |  |  |  |  |
| Satellite data | May 30 | Jul 8 | Jul 28 | Sep 27 | Oct 7 | Oct 5 |
| All-model spread | 33 | 17 | 13 | 35 | 21 | 32 |
|  |  |  |  |  |  |  |
| CanESM5 spread* | 5 | 4 | 4 | 6 | 5 | 9 |
| IPSL-CM6A-LR spread* | 6 | 8 | 5 | 10 | 9 | 14 |
| CESM LE spread* | 3 | 4 | 5 | 6 | 5 | 5 |

**Table 3.** Pan-Arctic, satellite-era (1979–2014) medians of seasonal sea ice transition dates. The satellite-era medians and the all-model spreads are calculated using the first ensemble member from each model. Models labeled with * show the spread in medians between the first 30 ensemble members of that model. Model spreads are given in days and all metrics are calculated between 66-84.5°N.

We find that for all spring transition metrics, the model spread exceeds estimations of internal variability, which show a maximum of 6 days between ensemble members (Table 3). Of the spring sea ice transition dates (melt onset, opening and 215 break-up), the sea ice melt onset dates show the largest spread in satellite-era medians between the models (33 days) (Table 3). This range is skewed late by the CNRM-ESM2-1 and CNRM-CM6-1. If the two CNRM models are excluded, the spread in median melt onset dates is 19 days instead of 33 days, still larger than the other two spring metrics (17 days for opening and 13 days for break-up). As the CNRM melt onset dates are more than a week later than the other models, this also means that differences between the CNRM models and the other models are unlikely explained by internal variability alone. The CNRM 220 models are further discussed in Sect. 4.5. Of the spring transition dates, the internal variability is highest for the melt onset





dates, particularly in the marginal ice zones (Fig. 9). High variability between ensemble members in the marginal ice zones is likely related to the interannual variations in the position of the ice edge. Additionally, modeled melt onset is defined using daily surface temperature, which exhibits greater variability than daily ice concentration (Smith and Jahn, 2019).

## 4.2 Fall transitions

**Figure 6.** Freeze onset dates averaged over 1979–2014 at each grid cell using satellite data (a), the first ensemble member of the CESM LE (b) and the first ensemble member of each CMIP6 model (c-q). Stippling indicates where freeze onset dates exist in less than 20% of years in the time range. Models on tripolar grids produce plot gaps filled by gray lines.

In the satellite data, the median freeze onset date is September 27 and the median freeze-up date is October 7. In the models, the satellite-era medians of freeze onset fall between September 28–November 2 and median freeze-up dates fall between October 8–October 29. Models therefore tend to show later freeze onset than observed (Figs. 6-8). Only five of the



**Figure 7.** Freeze-up dates (15% ice concentration threshold) averaged over 1979–2014 at each grid cell using satellite data (a), the first ensemble member of the CESM LE (b) and the first ensemble member of each CMIP6 model (c-q). Stippling indicates where freeze-up dates exist in less than 20% of years in the time range. Models on tripolar grids produce plot gaps filled by gray lines.

sixteen models fall within the maximum range of internal variability (10 days) of the satellite data (Table 3). Freeze-up is also generally delayed in models compared to satellite data, with seven of the sixteen models falling within the maximum range of internal variability (9 days) of the observations. The observed time between median freeze onset and freeze-up (the freeze period) is ten days, and this is twenty-nine days shorter than the time between melt onset and opening (the melt period) (Table 3).

The sea ice closes when it passes the 80% ice concentration threshold, and the median closing date occurs on October 5 in satellite data and between October 4–November 5 in the models (Table 3). Ice freeze-up occurs before the date of closing by definition, as both are defined using ice concentration, but areas closer to the Central Arctic (that fall below 80% but not 15%)



**Figure 8.** Closing dates (ice passes the 80% ice concentration threshold) averaged over 1979–2014 at each grid cell using satellite data (a), the first ensemble member of the CESM LE (b) and the first ensemble member of each CMIP6 model (c-q). Stippling indicates where closing dates exist in less than 20% of years in the time range. Models on tripolar grids produce plot gaps filled by gray lines.

skew the median of the closing dates earlier. The median length of the seasonal gain-of-ice period, the time between freeze-up and closing, is 7 days in satellite data and 4–14 days in models. Thus the seasonal loss-of-ice period is almost three times as long as the seasonal gain-of-ice period.

Like the spring transition metrics, we find that Pan-Arctic model differences in fall transition metrics are unlikely due to internal variability alone. Of the fall sea ice transition dates (freeze onset, freeze-up and closing), freeze onset shows the largest spread in satellite-era median between models (35 days) (Table 3). Freeze-up and closing have median spreads between the models of 21 and 32 days respectively. The maximum average standard deviation between the ensemble members of CESM LE, CanESM5 and IPSL for the fall transition metrics is 14 days. Therefore the model spreads of all fall transition metrics



exceed estimations of internal variability. In the fall transition dates, the average standard deviation between ensemble members
is highest for marginal ice zone freeze onset dates (Fig. 9). As described for melt onset, this large internal variability is due to
the changing interannual position of the ice edge and the variability of surface temperature.

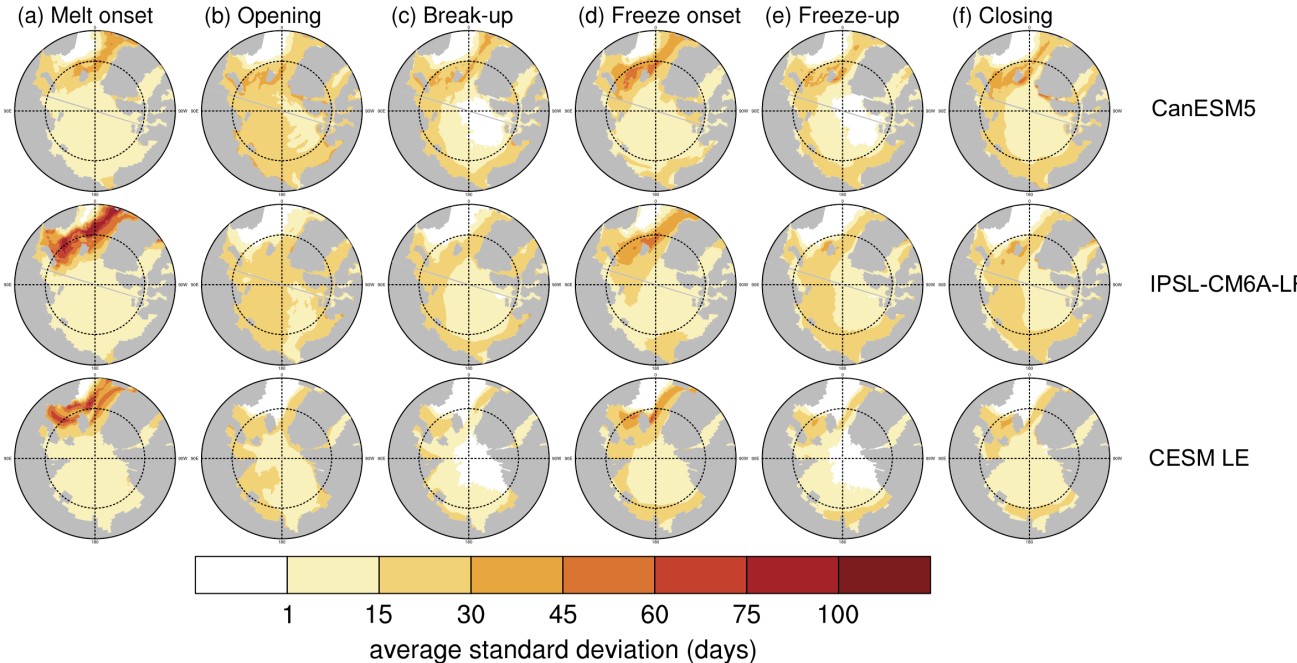

**Figure 9.** The average standard deviation between the first 30 ensemble members over 1979–2014 for (a) melt onset (b) opening (c) break-up
(d) freeze onset (e) freeze-up (f) closing. CanESM5 is displayed in the first row, IPSL is displayed in the second row and CESM LE is
displayed in the third row. The standard deviation is calculated at each grid cell for each year, and then the average of all years is plotted for
each grid cell. The same figure using all available ensemble members of each model is displayed in Supplementary Fig. S2.

The fall transition dates do not always occur in the expected order at each grid cell, with freeze up occurring at the same
time or earlier than freeze onset. In satellite data, simultaneous freeze onset and freeze-up dates may in part be explained
by the satellite retrieval algorithms: the PMW retrieval algorithm for freeze onset uses an ice concentration metric to derive
freeze onset at locations where the date can not be reliably derived using the weighted brightness temperature scheme (Markus
et al., 2009), but the data does not contain information on how often this back-up method is employed. Hence, the use of ice
concentration by both the freeze onset and freeze-up retrieval algorithms may contribute to cases where the dates are equal.

In models, the definitions of seasonal sea ice metrics aim to capture thermodynamic changes in the sea ice, but the similar
dates for freeze onset and freeze-up highlight that dynamic sea ice changes influence the ice concentration-based seasonal sea
ice transition metrics as well. While a particular grid cell may not register a persistent change in surface temperature below
the threshold for freeze onset (-1.8 °C), it is possible that the ice concentration of the grid cell surpasses 15% due to dynamic
transport into the grid cell, triggering the detection of freeze-up. In multiple models, freeze onset occurs later than freeze-up in





| | Melt season | Open water period (15%) | Outer ice-free period (80%) |
|---|---|---|---|
| ACCESS-CM2 | 121 | 88 | 77 |
| BCC-CSM2-MR | 136 | 87 | 77 |
| BCC-ESM1 | 124 | 78 | 72 |
| CanESM5 | 129 | 95 | 88 |
| CESM2 | 154 | 113 | 90 |
| CESM2-FV2 | 147 | 96 | 88 |
| CESM2-WACCM | 146 | 98 | 84 |
| CESM2-WACCM-FV2 | 130 | 77 | 76 |
| CNRM-ESM2-1 | 132 | 110 | 95 |
| CNRM-CM6-1 | 120 | 103 | 89 |
| EC-Earth3 | 127 | 85 | 80 |
| IPSL-CM6A-LR | 165 | 111 | 95 |
| MRI-ESM2-0 | 153 | 111 | 90 |
| NorESM2-LM | 148 | 100 | 82 |
| NorESM2-MM | 147 | 100 | 88 |
| CESM LE | 115 | 73 | 75 |
| | | | |
| Satellite data | 110 | 64 | 81 |
| All-model spread | 50 | 40 | 23 |
| | | | |
| CanESM5 spread* | 9 | 7 | 12 |
| IPSL-CM6A-LR spread* | 17 | 11 | 21 |
| CESM LE spread* | 9 | 9 | 9 |

**Table 4.** Lengths of pan-Arctic, satellite-era (1979–2014) medians of inter-seasonal transition periods in days. The satellite-era medians and the all-model spreads are calculated using the first ensemble member from each model. Models labeled with * show the spread in medians between the first 30 ensemble members of that model. Model spreads are given in days and all metrics are calculated between 66-84.5°N.

some parts of the Central Arctic. All models show freeze onset later than freeze-up occurring in the marginal ice zones, which are particularly susceptible to dynamic ice changes due to generally lower ice concentrations.

**4.3   Inter-seasonal transition periods**

Out of the three inter-seasonal periods of transition (the melt season, the open water period and the outer ice-free period), the outer ice-free period is the only one that is consistent with satellite data. The outer ice-free period (80% ice concentration thresholds) has an observed median length of 81 days and model medians falling between 72–95 days (Table 4).



In contrast, the melt season length and the open water period are too long in models compared to observations. Generally,
the greatest contribution to the differences between the observed and modeled open water period is from later than observed
freeze-up dates. The open water period has a median of 64 days in the satellite data and medians ranging between 73–113 days
in the models (a spread of 40 days) (Table 4). Likewise, modeled melt seasons that are too long compared to observations are
largely driven by later freeze onset dates. The observed median melt season length is 110 days and the model medians range
between 115-165 days (Table 4). Therefore the melt season length exhibits the largest model spread of all the inter-seasonal
periods (50 days). This is due to larger model ranges in both melt onset and freeze onset than the other transition dates, and the
contribution of each date to the model spread in melt season length is approximately equal (Table 4). The melt season length
model range is also skewed high by the IPSL model, which has a median melt season length of 165 days in its first ensemble
member. This is 11 days longer than the next longest model median, and the choice of ensemble member likely plays a role—
the IPSL model has a particularly large range of internal variability in the median melt season length (17 days compared to 9
days in the other two model sets) (Table 4).

## 4.4   Seasonal transitions affect sea ice area and thickness year-round

Model representations of seasonal sea ice transitions are expected to impact sea ice area and thickness because seasonal
transitions are strongly linked to the ice-albedo feedback (Perovich et al., 2008; Timmermans, 2015; Kashiwase et al., 2017;
Perovich, 2018; Lebrun et al., 2019). Ice loss earlier in the spring has been related to later ice gain in the fall (Stroeve et al.,
2014, 2016; Lebrun et al., 2019), and a weaker relationship has been described between later ice gain and earlier spring loss
during the following year (Lebrun et al., 2019). Both processes favor greater areas of open ocean for longer periods each year,
but little has been done to evaluate which transition metrics are most appropriate for describing pan-Arctic sea ice relationships.
Here we demonstrate year-round relationships using seasonal transition dates, March mean ice thickness and summer (June-
September) mean ice area. We show that pan-Arctic relationships between seasonal transitions and other ice characteristics are
most discernible using seasonal transition metrics with extensive spatial coverage (Fig. 10). Summer mean ice area is evaluated
instead of the ice area of a single month in order to better represent the integrated surface energy absorption as ice area declines.
Ice area and seasonal ice transition dates are practical for assessing sea ice in a pan-Arctic sense, as they are reliably available
for both models and observations. Discussion of the sea ice thickness here is limited to model projections, since observations
of Arctic sea ice thickness are temporally limited and contain large uncertainties (Bunzel et al., 2018).
We find that in satellite data, mean summer ice area (June–September) and the median timing of freeze onset are strongly
anti-correlated (R=-0.93) (Table 5 and Fig. 10). Lower summer ice area corresponds to a lower surface albedo, allowing for
greater shortwave absorption by the surface ocean and increasing ocean heat content (Timmermans, 2015), delaying the freeze
onset (Stroeve et al., 2014). Slightly weaker relationships exist between mean observed summer sea ice area and freeze-up
(R=-0.64) and closing (R=-0.81). In models, the greatest agreement on the correlation between mean summer ice area and fall
transition metrics is seen using the freeze onset dates, where all of the correlation coefficients that are statistically significant
at the 95% level (13 out of 16 models) are more negative than -0.73 (Table 5). Models tend to show later freeze onset than
observed, as discussed in Sect. 4.2, and despite this offset the observed relationship between summer ice area and freeze onset



**Figure 10.** Scatter plots of median seasonal sea ice transition metrics versus other ice characteristics for CMIP6 models (colors), CESM LE (gray) and satellite data (black). Each scatter point represents one year in one ensemble member from 1979-2014. Panels (a-d) show relationships with median opening and closing dates and panels (i-l) show relationships with median break-up and freeze-up dates. Metrics are scattered against mean summer (June–September) ice area in the first and fourth columns and March mean ice thickness in the second and third columns. All metrics are scattered against ice characteristics from the same year, except those in the second column, in which fall transition metrics are scattered against the next year's mean March ice thickness. All available ensemble members are used for CESM LE, CanESM5 and IPSL. All metrics are calculated between 66-84.5°N. All models are represented in each panel (a)-(l), but the labels are distributed across panels (h), (k) and (l).



|  | Melt onset | Opening (80%) | Break-up (15%) | Freeze onset | Freeze-up (15%) | Closing (80%) |
|---|---|---|---|---|---|---|
| ACCESS-CM2 | 0.66[a] | 0.78[a] | 0.22 | -0.84[a] | -0.77[a] | -0.73[a] |
| BCC-CSM2-MR | 0.47[a] | 0.64[a] | 0.39[a] | -0.80[a] | -0.60[a] | -0.74[a] |
| BCC-ESM1 | 0.53[a] | -0.14 | -0.05 | -0.67[a] | -0.53[a] | -0.45[a] |
| CESM2 | 0.37[a] | 0.87[a] | 0.43[a] | -0.87[a] | -0.78[a] | -0.87[a] |
| CESM2-FV2 | 0.70[a] | 0.89[a] | 0.21 | -0.90[a] | -0.75[a] | -0.82[a] |
| CESM2-WACCM | 0.62[a] | 0.85[a] | 0.22 | -0.86[a] | -0.68[a] | -0.79[a] |
| CESM2-WACCM-FV2 | 0.61[a] | 0.81[a] | 0.47[a] | -0.84[a] | -0.73[a] | -0.78[a] |
| CNRM-ESM2-1 | 0.08 | -0.32 | -0.25 | 0.24 | 0.14 | 0.14 |
| CNRM-CM6-1 | 0.13 | -0.19 | -0.15 | -0.07 | -0.04 | -0.12 |
| EC-Earth3 | 0.85[a] | 0.65[a] | 0.53[a] | -0.93[a] | -0.83[a] | -0.85[a] |
| MRI-ESM2-0 | 0.49[a] | 0.66[a] | -0.05 | -0.91[a] | -0.82[a] | -0.84[a] |
| NorESM2-LM | 0.55[a] | 0.69[a] | 0.11 | -0.73[a] | -0.51[a] | -0.64[a] |
| NorESM2-MM | 0.56[a] | 0.18 | -0.27 | -0.74[a] | -0.61[a] | -0.49[a] |
|  |  |  |  |  |  |  |
| CanESM5 | 0.73[a] | 0.65[a] | 0.11[a] | -0.83[a] | -0.68[a] | -0.71[a] |
| IPSL-CM6A-LR | 0.57[a] | 0.70[a] | 0.12[a] | -0.88[a] | -0.80[a] | -0.84[a] |
| CESM LE | 0.54[a] | 0.41[a] | 0.06[a] | -0.87[a] | -0.43[a] | -0.56[a] |
|  |  |  |  |  |  |  |
| Satellite data | 0.81[a] | 0.72[a] | 0.65[a] | -0.93[a] | -0.64[a] | -0.81[a] |

**Table 5.** Correlation coefficients (R-values) between seasonal sea ice transition dates and mean summer (June–September) sea ice area of the same year from 1979–2014. Values with [a] are statistically significant at the 95% confidence level. Correlation coefficients and p-values for models in the first thirteen rows are determined using one ensemble member, for CanESM5 using all 35 ensemble members, for IPSL using all 30 ensemble members and CESM LE using all 40 ensemble members. All values are calculated between 66-84.5°N.

is captured well by the models (Fig. 10). Summer ice area is generally larger in satellite data than in the models, but falls within the model spread. Relationships between fall transition dates and mean summer ice thickness are similar but slightly weaker

than found with ice area (Supplementary Table S3).

In the models, the timing of fall transition dates are strongly correlated with the March mean ice thickness (Table 6 and Fig. 10), but do not affect the March ice area of the following year (Supplementary Table S3). The differences in correlation coefficients indicate that increased heat absorption and delayed freeze onset reduce the March thickness of the ice but have a much smaller impact on the ice area. This supports past work on the Canada Basin, showing that anomalous solar heat

input (Perovich et al., 2008) reduced ice thickness over the winter of 2007-2008 by 25% (Timmermans, 2015). The strongest correlations between March ice thickness and the previous year's fall transition metrics are found between freeze onset and March ice thickness, with statistically significant correlation coefficients ranging between -0.54 and -0.92 in the models (Table 6). Because of sea ice thickness uncertainties discussed earlier (Bunzel et al., 2018), we are unable to confidently evaluate



whether model biases in freeze onset impact the simulated relationship between freeze onset and March mean ice thickness
compared to observations. With respect to the other fall transition metrics, we find that statistically significant correlations
between March ice thickness and freeze-up/closing (which are both based on ice concentration) are less consistent between
models, and generally stronger for the closing dates rather than freeze-up dates (Table 6).

| | Melt onset | Opening (80%) | Break-up (15%) | Freeze onset | Freeze-up (15%) | Closing (80%) |
|---|---|---|---|---|---|---|
| ACCESS-CM2 | 0.25 | 0.64[a] | 0.3 | -0.76[a] | -0.79[a] | -0.67[a] |
| BCC-CSM2-MR | 0.43[a] | 0.55[a] | 0.35[a] | -0.82[a] | -0.65[a] | -0.77[a] |
| BCC-ESM1 | 0.49[a] | -0.12 | 0.02 | -0.68[a] | -0.60[a] | -0.52[a] |
| CESM2 | 0.30 | 0.72[a] | 0.29 | -0.88[a] | -0.84[a] | -0.91[a] |
| CESM2-FV2 | 0.60[a] | 0.65[a] | 0.13 | -0.88[a] | -0.73[a] | -0.76[a] |
| CESM2-WACCM | 0.48[a] | 0.62[a] | -0.06 | -0.79[a] | -0.54[a] | -0.72[a] |
| CESM2-WACCM-FV2 | 0.48[a] | 0.65[a] | 0.38[a] | -0.81[a] | -0.70[a] | -0.74[a] |
| CNRM-ESM2-1 | 0.32 | -0.26 | -0.17 | -0.17 | -0.04 | -0.07 |
| CNRM-CM6-1 | 0.09 | -0.10 | -0.12 | -0.23 | -0.17 | -0.24 |
| EC-Earth3 | 0.75[a] | 0.57[a] | 0.49[a] | -0.92[a] | -0.85[a] | -0.79[a] |
| MRI-ESM2-0 | 0.43[a] | 0.42[a] | -0.13 | -0.84[a] | -0.83[a] | -0.82[a] |
| NorESM2-LM | 0.42[a] | 0.60[a] | 0.07 | -0.74[a] | -0.53[a] | -0.63[a] |
| NorESM2-MM | 0.38[a] | 0.01 | -0.37[a] | -0.54[a] | -0.41[a] | -0.37[a] |
| | | | | | | |
| CanESM5 | 0.64[a] | 0.51[a] | 0.01 | -0.73[a] | -0.64[a] | -0.64[a] |
| IPSL-CM6A-LR | 0.39[a] | 0.54[a] | 0.09[a] | -0.88[a] | -0.77[a] | -0.80[a] |
| CESM LE | 0.26[a] | 0.14[a] | -0.11[a] | -0.79[a] | -0.39[a] | -0.45[a] |

**Table 6.** Correlation coefficients (R-values) between seasonal sea ice transition dates and March sea ice thickness from 1979-2014. Spring
transition dates (melt onset, opening and break-up) are correlated with March mean ice thickness from the same year, while fall transition
dates (freeze onset, freeze-up and closing) are correlated with March mean ice thickness from the following year. Values with [a] are statistically
significant at the 95% confidence level. Correlation coefficients and p-values for models in the first thirteen rows are determined using one
ensemble member, for CanESM5 using all 35 ensemble members, for IPSL using all 30 ensemble members and CESM LE using all 40
ensemble members. All values are calculated between 66-84.5°N.

Modeled melt onset and opening dates both demonstrate weak to moderate relationships with the mean March ice thickness
of the same year (Table 6 and Fig. 10). Thinner March sea ice generally corresponds with earlier median melt onset and
opening dates. For March ice thickness and melt onset, statistically significant correlations range from 0.26-0.75, with the
CESM LE representing the weakest relationship in that range (Table 6). For March ice thickness and opening dates, statistically
significant correlations range from 0.14-0.65 (Table 6). One might expect that thinner ice would correspond to earlier break-
up dates, because thinner ice is easier to melt out or split apart. However, models do not agree on the sign or statistical
significance of any relationship between break-up (which are defined using ice concentration, like opening dates) and March



mean ice thickness. This lack of relationship is a strong indication that the spatial coverage of break-up dates is not sufficient for describing pan-Arctic sea ice feedbacks. Relationships between spring transition dates and March ice area are weaker than those between spring transition dates and March ice thickness (Supplementary Table S4).

Melt onset and opening are related to mean summer ice area in both satellite data and models (Table 5 and Fig. 10) (excluding the CNRM models, which are discussed in Sect. 4.5). Earlier melt is correlated with lower mean summer ice area with a correlation coefficient of 0.81 in satellite data and a range of 0.37-0.85 in statistically significant model correlations (Table 6). Earlier opening is slightly less correlated with lower mean summer ice area with a correlation coefficient of 0.72 in satellite data, and the models range between 0.41-0.89 in statistically significant correlations (Table 6). Both earlier melt onset and opening dates decrease the surface albedo—the former though the formation of melt ponds and the latter through the presence of more open ocean. This once again facilitates greater surface absorption, which has been shown to increase the ocean heat content and decrease the summer sea ice cover (Stroeve et al., 2014). Relationships also exist between melt onset/opening and summer sea ice thickness (Supplementary Table S3), but since the summer sea ice is already quite thin, greater ocean heat content is more likely to affect the ice area than it is in March, when ice is much thicker overall. Models do not agree on the sign or magnitude of the correlation between break-up and summer ice area, again indicating that the spatial coverage of break-up dates is insufficient for describing pan-Arctic sea ice processes.

## 4.5 Seasonal transitions can compensate for unrealistic sea ice characteristics

CNRM-CM6-1 and CNRM-ESM2-1 demonstrate that biases in seasonal sea ice transitions can unrealistically compensate for other sea ice biases. As mentioned in Sect. 4.1 and 4.4, the CNRM models show median melt onset dates occurring 10-14 days later than the next latest model and considerably later (14-18 days) than those found in satellite data (Table 4). The largest differences in melt onset between the CNRM models and both satellite data and the other models are found in the Central Arctic (Fig. 3k, l). While melt onset dates fall late in the CNRM models, their September ice areas are overall realistic (Voldoire et al., 2019) and fall within the spread of available models (Supplementary Fig. S1). The CNRM models are the only two models (out of sixteen) that the lack statistically significant correlations between later melt onset and larger summer ice area seen in most models and observations (Sect. 4.4). Furthermore, mean ice thickness in the CNRM models from 1979-2014 is too low (Supplementary Fig. S1). Thus, the models' ability to produce realistic September sea ice areas likely relies on the biased seasonal transition: late melt onset acts to retain thin ice that would otherwise be lost over the summer by shortening the length of the melt season. Following melt onset, the CNRM models have median opening and break-up dates that fall fully within the model spread, indicating that the impact of seasonal transition biases can be be large, even if the biases exist only in one metric.

The cause of delayed melt onset in CNRM models is not currently clear. Melt onset is the only transition metric that captures changes at the surface of the snowpack rather than a change in ice concentration. Recent work suggests that the winter snow on the sea ice is too thick in the CNRM models (Voldoire et al., 2019), over-insulating the sea ice and preventing it from reaching realistic ice thicknesses (Voldoire et al., 2019). We find that the over-insulation in CNRM models may be more related to September–November snow thickness, since the CNRM models show the largest area of 15-30 cm deep snow of





all the models across this time frame (Supplementary Fig. S3), but show similar snow thicknesses compared to other models
during December–February (Supplementary Fig. S4). Delayed melt onset could also be related to the use of the GELATO
sea ice model, as the CNRM models are the only models used in this study that use the GELATO model (Supplementary
Table S2). Since GELATO has a single snow-on-sea-ice layer and fixed albedos for dry snow and melting snow (0.88 and 0.77
respectively) (Voldoire et al., 2019), simplified processes in GELATO may contribute to snow biases.

## 5  Conclusions

Seasonal sea ice transitions can be characterized by various metrics (melt onset, opening, break-up, freeze onset, freeze-up
and closing), and each metric represents a distinct stage of sea ice loss or gain. As such, seasonal transitions provide unique
insights into Arctic sea ice processes, but they have so far been under-utilized in evaluating climate models due to a lack of
long-term observational products and daily model output, as well as the complexities of defining seasonal transitions. Taking
advantage of newly available daily model output (Notz et al., 2016) and observational data of seasonal transitions (Steele et al.,
2019), we show that models capture the observed asymmetry in seasonal sea ice transitions, with spring ice loss taking about
1.5–2 months longer than fall ice growth (Figs. 3–8). Models also generally agree with satellite data on the timing of spring
transitions, but eleven out of sixteen models show median freeze onset dates later than observed, such that the differences
between each model's median freeze onset date and the observed date exceed the largest estimations of internal variability
(Table 3). Likewise, in almost half of the models (seven out of sixteen), the difference between the median freeze-up date and
the observed date exceed the largest estimations of internal variability. Delayed freeze onset and freeze-up extend simulated
melt seasons and open water periods respectively, making the outer ice-free period (the time between ice opening and closing)
the only inter-seasonal period in which models consistently agree with satellite observations.

We find that differences in seasonal transitions between models are unlikely due to internal variability alone, and are hence
likely a reflection of model differences. Sea ice metrics are each impacted differently by internal variability: models do not
agree on a metric most affected, and no single model exhibits the greatest internal variability across all metrics. Despite
the uncertainty associated with internal variability, all metrics show pan-Arctic model spreads exceeding even the largest
estimations of internal variability in seasonal sea ice transition metrics (Tables 3 and 4). The largest standard deviations between
ensemble members are seen in the inflow regions of melt and freeze onset dates (Fig. 9), and this is due to the changing
interannual position of the ice edge and the variability of surface temperature.

Because differences in seasonal sea ice transition metrics between models are unlikely due only to internal variability, these
metrics can be used for evaluating differences between models in terms of other sea ice characteristics. We show that pan-Arctic
relationships between transition metrics and sea ice area and thickness depend on the spatial coverage of the metric (Fig. 10).
Out of the six transition dates, melt and freeze onset dates consistently cover the largest area of the Arctic, and they are most
closely related to pan-Arctic ice area and mean thickness. Low mean summer ice area delays freeze onset (Table 5), which in
turn leads to lower March ice thickness (Table 6). Thinner March ice leads to earlier melt onset and, again, low mean summer
ice area (Table 5). Other relationships between sea ice area and thickness are somewhat discernible using opening and closing

dates, but almost indistinguishable using break-up and freeze-up dates (Fig. 10). Since the differences in relationship strengths are seen across definitions that use both surface temperature and ice concentration-based definitions, these differences are more likely related to the spatial coverage of the seasonal sea ice transition dates rather than their defining variables (Tables 5 and 6, Fig. 10). While models tend to show later freeze onset than observed, this offset does not impact the ability of the models to produce the observed relationship between lower summer ice area and later freeze onset.

Finally, we demonstrate how seasonal sea ice transition metrics can provide context to sea ice changes that otherwise lack quantified explanations. We find that CNRM-ESM2-1 and CNRM-CM6-1 exhibit biases in both melt onset (late) and ice thickness (thin), exemplifying how seasonal ice transitions can compensate for other unrealistic aspects of the sea ice. Late melt onset helps retain thin ice throughout the summer such that both CNRM models exhibit realistic September sea ice areas for the wrong reasons. Seasonal sea ice transitions metrics therefore provide a process-based constraint on model simulations in addition to the commonly used September and March sea ice areas (Stroeve et al., 2012; Rosenblum and Eisenman, 2017).

To conclude, routinely saved daily sea ice variable output (in particular sea ice concentration and surface temperature) will be critical for using seasonal transitions as a new metric to assess and quantify model uncertainties associated with Arctic sea ice simulations. Since a new observational data product for these seasonal sea ice transition now exists (Steele et al., 2019), seasonal sea ice transition dates should be used routinely in the future to better identify model biases in sea ice evolution as well as the sources of these biases.

*Data availability.* CMIP6 data are publicly available at the World Climate Research Programme CMIP6, supported by the Department of Energy's Lawrence Livermore National Laboratory and the Earth System Grid Federation (https://esgf-node.llnl.gov/projects/cmip6/, last access, 19 February 2020). All CMIP6 model output used is cited in the reference list. CESM LE data are publicly available at the National Center for Atmospheric Research Climate Data Gateway (https://www.earthsystemgrid.org/, last access: 19 February 2020). Version 1 of the Arctic Sea Ice Seasonal Change and Melt/Freeze Climate Indicators from Satellite Data are publicly available at the National Snow and Ice Data Center (https://nsidc.org/data/NSIDC-0747/versions/1, last access: 26 Aug 2019).

*Author contributions.* AS and AJ conceived the study, and AS analyzed the data and prepared the manuscript, with guidance and edits from AJ and MW.

*Competing interests.* The authors declare that they have no conflict of interest.

*Acknowledgements.* A. Smith's contribution is supported by the Future Investigators in Earth System Science Grant no. 80NSSC19K1324, the National Science Foundation Graduate Research Fellowship grant no. DGE 1144083, and NSF-OPP award 1847398. A. Jahn acknowledges support from NSF-OPP award 1847398. M. Wang is supported by NSF grant 1751363. She is also funded by the Joint Institute for



the Study of the Atmosphere and Ocean (JISAO) under NOAA Cooperative Agreement NA15OAR4320063, Contribution No. 2020-1056 and by NOAA Arctic Research Program, Pacific Marine Environmental Laboratory contribution number 5076. We acknowledge the World Climate Research Programme, which, through its Working Group on Coupled Modelling, coordinated and promoted CMIP6. We thank the climate modeling groups for producing and making available their model output, the Earth System Grid Federation (ESGF) for archiving the data and providing access, and the multiple funding agencies who support CMIP6 and ESGF. The CESM project is supported by the

National Science Foundation and the Office of Science (BER) of the U.S. Department of Energy. Computing resources for the CESM LE were provided by the Climate Simulation Laboratory at NCAR's Computational and Information Systems Laboratory (CISL), sponsored by the National Science Foundation and other agencies. Five of the CESM LE simulations were produced at the University of Toronto under the supervision of Paul Kushner. NCL (2017) was used for data analysis.



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
