# Peer review of "Seasonal transition dates can reveal biases in Arctic sea ice simulations"

_The Cryosphere, 2020_

## Referee Comment (RC1) · Anonymous Referee #1 · 2 May 2020

In this study, the authors evaluate climate model performance for describing the Arctic sea ice seasonal cycle with a series of metrics that describe stages of the melt and freeze-up periods. Modeled sea ice concentrations and surface temperatures are used to approximate significant dates of the melt and freeze cycle obtained from passive microwave satellite observations. The authors find that the models capture a similar asymmetry in the melt/freeze cycle where the melting period is longer than the corresponding freeze-up period as seen in observations. Overall, the models generally agree with observations on the timing of spring melt, but several show delays in the timing of the freeze-up period relative to the observations. The differences between models for these variables exceed expected differences due to internal variability of the model, thus, the authors propose using the seasonal sea ice transition metrics to

evaluate climate model performance. I find that the paper is very well written, interesting, and thorough in reporting the results of the study. I have a few very minor comments that the authors should address before publication as outlined below.

Minor Comments

L 205-206: I'm not sure exactly what you mean by the model spread shifting "earlier toward the satellite data". Can you please expand or rephrase this in the text?

L 206: Inflow regions are not specifically defined anywhere in the paper. It would be worth it to state where these are (e.g., Bering Strait, etc.).

L 249: It would be beneficial to expand a bit on the ice concentration metric used by Markus et al. (2009) when the algorithm does not detect a clear freeze onset signal from the brightness temperatures. Specifically, that the threshold used by Markus is 80% SIC, which in theory makes some unknown quantity of the satellite freeze onset dates more comparable to the closing dates, than freeze-up dates. This is likely contributing to the instances where the freeze transition dates are out of order (e.g. as you state in lines 247-248).

Figures 3-8: Since Jan – Mar are repeated twice in the color scale, it would be easier for readers to see that the repeated dates in the blue colors are for the following year if this was denoted on the scale markings somehow.

Technical Corrections

L 30: Typo – Melt ponds _decrease_ the albedo of the surface

L 328: Typo – the former _through_ the formation of. . .

---

## Referee Comment (RC2) · Anonymous Referee #2 · 8 May 2020

General Comments

The manuscript suggests to evaluate CMIP6 sea ice simulations using ice seasonality metrics. The authors define a set of seasonal metrics based on sea ice concentration and surface temperature simulated by CMIP6 and CESM earth system models. A recent satellite database describing several ice seasonality metrics is also used to evaluate models.

A first part is focused on transition period between the different melt and freezing dates. They find an asymmetry between spring ice loss and fall ice growth in both satellite observations and model simulations. They also show that most models have a late freeze onset compared to observations.

[Figure]

In the second part the correlation between seasonal metrics and sea ice area and thickness is studied. The authors find a good correlation between freezing dates , sea ice area and thickness. These correlations allow to highlight sea ice biases in some models which are compensated by other processes. The authors give the example of CNRM model which has biases in melting and in sea ice thickness but which represent realistic sea ice area for the wrong reasons.

I found this paper very interesting. It includes newly observation database and suggests an interesting and novel approach to evaluate earth system models sea ice simulations. I stress the good work the authors made by analysing a large set of earth system models with several members besides satellite observations. I think that this work can make a great contribution to the literature after following comments are addressed in the context of a minor revision.

Specific Comments

P1 line 13 "the spread between climate model projections of sea ice has been on the order of millions of square kilometers in Coupled Model Intercomparison Project (CMIP)" . Can you specify about which parameter you are talking ? Sea ice coverage ?

P4 line 92 "(select figures using all available members are provided in the Supplement)." : Can you specify which figures ?

P9 to P17 : I think it would be beneficial to add maps of the four intra-seasonal periods (melt , freeze , seasonal loss-of-ice and seasonal gain-of-ice periods) and a table of spatial median ( and standard deviation) for each model and observation as for table 4. Moreover, you look at the difference between the spatial of the metrics to describe the median of the intra-seasonal periods. But as median(A) - median (B) $\neq$ med(A -

B), calculating the intra-seasonal period for each pixel before doing the spatial median seems more appropriate.

P11 lines 205-207 "the model spread (May 15- June 3) " : Is it really June 3 or is it June 13 here?

P14 line 228 : What do you mean by "internal variability of the satellite data" ?

p20 line 302 " (Supplementary Table S3)" : I guess you mean Table S4.

p22 line 320 "This lack of relationship is a strong indication that the spatial coverage of break-up dates is not sufficient for describing pan-Arctic sea ice feedbacks. " : I wonder if the lack of relationship between March mean ice thickness and break up date can be explained by the inverse relation between ice growth and thickness which explains that the thinner the ice, the more efficient the growth. This relation should temper the delay in break up ( see Bitz & Roe, 2004 and Lebrun et al. 2019 ).

Bitz, C. M., Holland, M. M., Hunke, E. C. and Moritz, R. E.: Maintenance of the Sea-Ice Edge, J. Climate, 18(15), 2903?2921, doi:10.1175/JCLI3428.1, 2005.

Lebrun, M., Vancoppenolle, M., Madec, G. and Massonnet, F.: Arctic sea-ice-free season projected to extend into autumn, The Cryosphere, 13(1), 79?96, doi:https://doi.org/10.5194/tc-13-79-2019, 2019.

p22 line 247 " indicating that the impact of seasonal transition biases can be be large" : you should remove a "be"

[Figure]

Figure 2 : This figure seems not describe in detail in the main text. You should move it in supplementary.

Figures 3 and 6: Can you remind the definition criteria for melt and freeze onset dates in both caption as you did for fig 4,5,7 and 8 ?

Table 3 : What do you mean by "spread" ?

Table 3 - Table 4 : Can you also add a spatial standard deviation for each model and observation?

Table 6 or Table S4 : Caption for both tables are exactly the same. I guess it is a mistake you should fix.

---

## Author Comment (AC1) · 6 Jun 2020

We thank the referee very much for their help and constructive comments. Responses to the minor comments and technical corrections are below.

Minor Comments

1. L 205-206: I'm not sure exactly what you mean by the model spread shifting "earlier toward the satellite data". Can you please expand or rephrase this in the text?

Yes, we will rephrase this sentence to clarify that the model data becomes more similar to the satellite data in terms of the median melt onset dates when excluding the CNRM models.

2. L 206: Inflow regions are not specifically defined anywhere in the paper. It would be worth it to state where these are (e.g., Bering Strait, etc.).

Thank you for this suggestion. We will define "inflow regions" in the manuscript.

3. L 249: It would be beneficial to expand a bit on the ice concentration metric used by Markus et al. (2009) when the algorithm does not detect a clear freeze onset signal from the brightness temperatures. Specifically, that the threshold used by Markus is 80% SIC, which in theory makes some unknown quantity of the satellite freeze onset dates more comparable to the closing dates, than freeze-up dates. This is likely contributing to the instances where the freeze transition dates are out of order (e.g. as you state in lines 247-248).

We agree that expanding on the Markus et al. (2009) algorithm in this section would be beneficial. We will add text to the manuscript explaining the 80% SIC threshold and how this likely contributes to the fall transition dates that occur out of the expected order.

4. Figures 3-8: Since Jan – Mar are repeated twice in the color scale, it would be easier for readers to see that the repeated dates in the blue colors are for the following year if this was denoted on the scale markings somehow.

We agree and will adjust Figures 3-8 to clearly denote that the second January, February and March on the color bar occur in the following year.

Technical Corrections

1. L 30: Typo – Melt ponds _decrease_ the albedo of the surface

We will correct this typo.

2. L 328: Typo – the former _through_ the formation of...

We will correct this typo.

---

## Author Comment (AC2) · 6 Jun 2020

We thank the referee very much for their help and constructive comments. Responses to the specific comments are below.

Specific Comments

1. P1 line 13 "the spread between climate model projections of sea ice has been on the order of millions of square kilometers in Coupled Model Intercomparison Project (CMIP)" . Can you specify about which parameter you are talking? Sea ice coverage ?

Yes, this should read "the spread between climate model projections of sea ice AREA has been on the order of millions of square kilometers in Coupled Model Intercomparison Project (CMIP)" and we will correct this in the manuscript.

[Figure]

2. P4 line 92 "(select figures using all available members are provided in the Supplement)." Can you specify which figures ?

Yes, here we are referring to the plots in Figure 9, which are reproduced using all available members in Figure S2. We will clarify this in the text.

3. P9 to P17 : I think it would be beneficial to add maps of the four intra-seasonal periods (melt , freeze , seasonal loss-of-ice and seasonal gain-of-ice periods) and a table of spatial median ( and standard deviation) for each model and observation as for table 4. Moreover, you look at the difference between the spatial of the metrics to describe the median of the intra-seasonal periods. But as median(A) - median(B) /= med(A-B), calculating the intra-seasonal period for each pixel before doing the spatial median seems more appropriate.

Thank you for this suggestion. The maps and median tables were initially excluded from the manuscript due to concerns related to the length of the manuscript and the number of figures. Additionally, plots and tables related to the intra-seasonal periods can be difficult to interpret since they show negative values when the dates fall out of order. We will recreate these plots and tables to reassess their utility. We will also calculate the spatial standard deviations and evaluate whether they add to the content of the manuscript.

With respect to the median differences, we are calculating the intra-seasonal period for each pixel before doing the spatial median, so any in-text discussion of the intra-seasonal periods refers to values calculated in this way. We will add text to the manuscript to clarify this process.

4. P11 lines 205-207 "the model spread (May 15- June 3) " : Is it really June 3 or is it June 13 here?

Excluding the CNRM models, the latest satellite-era median melt onset date occurs in the CanESM5 on June 3. We will move the text "(May 15 - Jun 3)" to the end of the

sentence to make this specification clearer.

5. P14 line 228 : What do you mean by "internal variability of the satellite data" ?

Here we are describing how many of the models fall within 10 days of the satellite data in terms of their satellite-era median freeze onset dates. The value of 10 days is found in Table 3, which shows the spread (latest minus earliest) in medians between the first thirty ensemble members from individual models. Of these models, the largest estimation of internal variability in freeze onset dates is 10 days. We will clarify this assessment in the text.

6. p20 line 302 " (Supplementary Table S3)" : I guess you mean Table S4.

Yes, thank you. We will change the text to read "(Supplementary Table S4)".

7. p22 line 320 "This lack of relationship is a strong indication that the spatial coverage of break-up dates is not sufficient for describing pan-Arctic sea ice feedbacks. " : I wonder if the lack of relationship between March mean ice thickness and break up date can be explained by the inverse relation between ice growth and thickness which explains that the thinner the ice, the more efficient the growth. This relation should temper the delay in break up ( see Bitz & Roe, 2004 and Lebrun et al. 2019 ).

Bitz, C. M., Holland, M. M., Hunke, E. C. and Moritz, R. E.: Maintenance of the Sea-IceEdge, J. Climate, 18(15), 2903?2921, doi:10.1175/JCLI3428.1, 2005.

Lebrun, M., Vancoppenolle, M., Madec, G. and Massonnet, F.: Arctic sea-ice-free season projected to extend into autumn, The Cryosphere, 13(1), 79-96, doi:https://doi.org/10.5194/tc-13-79-2019, 2019.

Thank you for this suggestion. We will evaluate the relationship between March mean ice thickness and break-up in the context of the suggested papers, and we will add related text to this section of the manuscript.

8. p22 line 247 " indicating that the impact of seasonal transition biases can be be

large" you should remove a "be"

We will correct this typo.

9. Figure 2 : This figure seems not describe in detail in the main text. You should move it in supplementary.

We agree and will move Figure 2 to the Supplement.

10. Figures 3 and 6: Can you remind the definition criteria for melt and freeze onset dates in both caption as you did for fig 4,5,7 and 8 ?

Yes, thank you for this suggestion. We will add a short phrase to the captions of Figures 3 and 6 describing how the melt and freeze onset dates are derived.

11. Table 3 : What do you mean by "spread" ?

Here the word "spread" is referring to the difference between the earliest and the latest dates found using the first thirty members of each model. We will define "spread" in the text for clarity.

12. Table 3 - Table 4 : Can you also add a spatial standard deviation for each model and observation?

We will calculate the spatial standard deviations and evaluate whether they add to the content of the manuscript.

13. Table 6 or Table S4 : Caption for both tables are exactly the same. I guess it is a mistake you should fix.

Table 6 shows correlation coefficients between seasonal sea ice transition dates and March sea ice thickness from 1979-2014 while Table S4 shows correlation coefficients between seasonal sea ice transition dates and March sea ice area over the same time period. To clarify what is different for figures and tables with very similar captions, we will add "As in Table 6, Table S4 shows...but for..."

---

## Author Response (AR1)

**Revision of "Seasonal transition dates can reveal biases in Arctic sea ice simulations"**

We thank the two referees and the editor for their constructive and positive feedback on the discussion paper. Below, we have included descriptions of changes made due to the detection of two coding bugs in the process of making revisions (which do not change any conclusions), followed by point-by-point replies to the two reviews.

**Description of two corrections made in the revised manuscript**

*1. Satellite-era medians versus means*
In the initial submission of the manuscript, a metric called the "satellite-era median" was defined to represent a measure-of-center of the average spatial distribution of each seasonal transition date from 1979-2014. This process was described in the Methods section:

> "To quantitatively compare the models to each other and to observations in a pan-Arctic sense, we take the median of the resulting distribution, shown as a histogram in Fig. 2. This value is referred to as the "satellite-era median".

When addressing the comments of the referees, code bugs were found associated with the medians, so that what was described as medians in the original submission were actually means, and the means of the satellite data and the means for models with more than 30 ensemble members were calculated from histograms of different bin sizes than the CMIP models with less members. We have analyzed the impacts of these code bugs, and have found that they do not change any of the conclusions.

Using the means is not more or less valid than using the medians, as both still achieve the stated original goal: they represent a valid measure-of-center that accounts for differences in spatial coverage. However when using mean values, it is more accurate to use area-weighted spatial means, since means from the distributions lose decimal places in the binning process (otherwise they are equivalent). **As the conclusions are not impacted and both are valid metrics, we have chosen to continue to show the means (now calculated as area-weighted spatial means for accuracy) and changed the Methods to accurately reflect this, which reduced the changes compared to the discussion paper.**

For reference, details of the effects of using area-weighted spatial means (referred to from here on as "means") versus medians from the areal distributions (referred to from here on as "medians") are described below, based on the detailed analysis we have performed. **We have also edited the Methods section to briefly describe how using medians instead of means affects the results.**

*Detailed description of the impact of using means instead of medians:*
Due to the skew of the distributions (Figure R1, which was included as Figure 2 in the original submission and is now Figure S3 in the Supplement in response to Referee #2), using mean values results in earlier spring transition dates and later fall transition dates compared to using median values. In the models, the average difference between the mean and median for all transition dates is 7 days and the maximum difference is 19 days (associated with freeze onset). In the satellite data, the average difference is also 7 days and the maximum difference is 16 days (also associated with freeze onset). To demonstrate an example of these differences, Tables R1 and R2 below show the median and mean melt onset and freeze onset days for each model and the number of days between them.

None of the conclusions are affected by the use of the mean versus the median. However, if we were to use the medians, two findings would be less-pronounced than reported based on the means.

1) **Melt onset in the CNRM models is later than in other models and satellite data:** The difference between the melt onset in the CNRM models and other models and satellite observations is smaller when medians are compared (see Table R1). However, as all models except the CNRM models and the CESM LE still fall earlier than or on the same day as the satellite data when medians instead of means are used, this does not affect the overall validity of this finding.

**2) Internal variability cannot explain the later simulated freeze onset and freeze-up:** Using medians, many models still demonstrate later freeze onset and freeze-up dates than observed, but fewer models fall outside of the maximum range of internal variability compared to the satellite data (Table R3). In turn, differences between modeled and observed melt seasons and open water periods are smaller as well. However, this does not affect our interpretation of the results since the models still tend to fall later than the satellite data in terms of freeze onset and freeze-up and longer in terms of melt season and open water period--the differences are just less pronounced and may more often be due to internal variability.

The other results in the manuscript are even less affected by using medians instead of means. The spreads between ensemble members in models with more than 30 members vary by an average of two days. The scatter plots presented in Figure 10 still maintain the same patterns, and the average differences in correlation coefficients for Tables 3 and 4 are of magnitude 0.01.

[Figure]

**Figure R1.** Area distributions of the average of each metric from 1979-2014: (a) melt onset (b) opening (c) break-up (d) freeze onset (e) freeze-up and (f) closing. Metrics are averaged from 66-84.5 N for satellite data (filled gray) and the first ensemble member of each model (all other colors). All models and satellite data are represented in each panel (a)-(f), but the color labels are distributed across panels (a)-(c).

| | Mean melt onset date | Median melt onset date | Difference in days (mean minus median) |
|---|---|---|---|
| ACCESS-CM2 | 3-Jun | 5-Jun | -2 |
| BCC-CSM2-MR | 24-May | 29-May | -5 |
| BCC-ESM1 | 30-May | 2-Jun | -3 |
| CanESM5 | 3-Jun | 6-Jun | -3 |
| CESM2 | 20-May | 28-May | -8 |
| CESM2-FV2 | 22-May | 27-May | -5 |
| CESM2-WACCM | 23-May | 29-May | -6 |
| CESM2-WACCM-FV2 | 21-May | 28-May | -7 |
| CNRM-ESM2-1 | 14-Jun | 18-Jun | -4 |
| CNRM-CM6-1 | 18-Jun | 21-Jun | -3 |
| EC-Earth3 | 2-Jun | 10-Jun | -8 |
| IPSL-CM6A-LR | 15-May | 30-May | -15 |
| MRI-ESM2-0 | 22-May | 27-May | -5 |
| NorESM2-LM | 21-May | 26-May | -5 |
| NorESM2-MM | 29-May | 3-Jun | -5 |
| CESM LE | 29-May | 16-Jun | -18 |
| Satellite data | 6-Jun | 10-Jun | -4 |

Table R1. Pan-Arctic, satellite-era (1979–2014) mean melt onset dates, median melt onset dates and the difference between them in days for the first ensemble member of each model and satellite data. This table has been adapted from Table 3 in the original manuscript.

| | Mean freeze onset date | Median freeze onset date | Difference in days (mean minus median) |
|---|---|---|---|
| ACCESS-CM2 | 6-Oct | 17-Sep | 19 |
| BCC-CSM2-MR | 8-Oct | 27-Sep | 11 |
| BCC-ESM1 | 2-Oct | 22-Sep | 10 |
| CanESM5 | 16-Oct | 29-Sep | 17 |
| CESM2 | 23-Oct | 10-Oct | 13 |
| CESM2-FV2 | 18-Oct | 3-Oct | 15 |
| CESM2-WACCM | 17-Oct | 2-Oct | 15 |
| CESM2-WACCM-FV2 | 17-Oct | 2-Oct | 15 |
| CNRM-ESM2-1 | 28-Oct | 18-Oct | 10 |

| | | | |
|---|---|---|---|
| CNRM-CM6-1 | 21-Oct | 8-Oct | 13 |
| EC-Earth3 | 10-Oct | 28-Sep | 12 |
| IPSL-CM6A-LR | 3-Nov | 20-Oct | 14 |
| MRI-ESM2-0 | 25-Oct | 11-Oct | 14 |
| NorESM2-LM | 17-Oct | 3-Oct | 14 |
| NorESM2-MM | 8-Oct | 22-Sep | 16 |
| CESM LE | 7-Oct | 24-Sep | 13 |
| Satellite data | 4-Oct | 18-Sep | 16 |

**Table R2.** Pan-Arctic, satellite-era (1979–2014) mean freeze onset dates, median freeze onset dates and the difference between them in days for the first ensemble member of each model and satellite data. This table has been adapted from Table 3 in the original manuscript.

| | Freeze onset | | Freeze-up | |
|---|---|---|---|---|
| | Number of models later than satellite data | Number of models outside greatest estimated range of internal variability | Number of models later than satellite data | Number of models outside greatest estimated range of internal variability |
| **Means** | 15 | 10 | 11 | 5 |
| **Medians** | 15 | 5 | 13 | 2 |

**Table R3.** The number of models out of sixteen that show later freeze onset and freeze-up dates in terms of their medians and means. This table also shows the number of models out of sixteen that are later than the satellite data by at least the largest number of days estimated to represent internal variability.

*2. Correction in Supplementary Table S4*

The correlation coefficients in the columns for freeze onset, freeze-up and closing in Supplementary Table S4 were meant to portray the relationship between the fall transition metrics and the March ice area of the following year, but by accident the correlation coefficients shown were for the same year (instead of the following year). This does not impact the conclusions, as differences in correlation coefficients for the third through sixth columns of Table S4 are of magnitude 0.01. The values in Table S4 have now been corrected.

**Point by point reply to Review comments:**
**Anonymous Referee #1**

In this study, the authors evaluate climate model performance for describing the Arctic sea ice seasonal cycle with a series of metrics that describe stages of the melt and freeze-up periods. Modeled sea ice concentrations and surface temperatures are used to approximate significant dates of the melt and freeze cycle obtained from passive microwave satellite observations. The authors find that the models capture a similar asymmetry in the melt/freeze cycle where the melting period is longer than the corresponding freeze-up period as seen in observations. Overall, the models generally agree with observations on the timing of spring melt, but several show delays in the timing of the freeze-up period relative to the observations. The differences between models for these variables exceed expected differences due to internal variability of the model, thus, the authors propose using the seasonal sea ice transition metrics to evaluate climate model performance. I find that the paper is very well written, interesting, and thorough in reporting the results of the study. I have a few very minor comments that the authors should address before publication as outlined below.

> We thank the referee very much for their constructive comments. We have made all the suggested changes (details below).

*Minor Comments*

1. L 205-206: I'm not sure exactly what you mean by the model spread shifting "earlier toward the satellite data". Can you please expand or rephrase this in the text?

> To clarify this, we have edited this sentence. It now reads: "Excluding the CNRM models (which show particularly late mean melt onset dates and are explored further in Sect. 4.5), the model spread (May 15--June 3) shifts earlier, and the mean melt onset dates from the remaining models all occur earlier than the satellite data."

2. L 206: Inflow regions are not specifically defined anywhere in the paper. It would be worth it to state where these are (e.g., Bering Strait, etc.).

> As requested, we have now added a definition of the "inflow regions" as the Chukchi Sea, Barents Sea and Greenland Sea, and of the "Atlantic inflow regions" as the Barents Sea and Greenland Sea.

3. L 249: It would be beneficial to expand a bit on the ice concentration metric used by Markus et al. (2009) when the algorithm does not detect a clear freeze onset signal from the brightness temperatures. Specifically, that the threshold used by Markus is 80% SIC, which in theory makes some unknown quantity of the satellite freeze onset dates more comparable to the closing dates, than freeze-up dates. This is likely contributing to the instances where the freeze transition dates are out of order (e.g. as you state in lines 247-248).

> As suggested, we have now expanded this discussion of the back-up sea ice concentration metric used by Markus et al. (2009): "In satellite data, simultaneous freeze onset and freeze-up dates may in part be explained by the satellite retrieval algorithms: the PMW retrieval algorithm for freeze onset uses an 80% ice concentration metric to derive freeze onset at locations where the date can not be reliably derived using the weighted brightness temperature scheme (Markus et al., 2009). This would skew the freeze

onset dates later and make them more similar to the closing dates. Hence, the use of ice concentration by both the freeze onset and freeze-up retrieval algorithms may contribute to cases where the dates are not sequential. A detailed assessment of this is not possible, however, as the data does not contain information on how often this back-up method is employed."

4. Figures 3-8: Since Jan – Mar are repeated twice in the color scale, it would be easier for readers to see that the repeated dates in the blue colors are for the following year if this was denoted on the scale markings somehow.

We agree and have adjusted Figures 2-7 such that the two Januarys are labeled with "year" and "year+1". We have also bolded the line on the color bar denoting January$_{year+1}$.

*Technical Corrections*

1. L 30: Typo – Melt ponds _decrease_ the albedo of the surface

Corrected.

2. L 328: Typo – the former _through_ the formation of...

Corrected.

**Anonymous Referee #2**

The manuscript suggests to evaluate CMIP6 sea ice simulations using ice seasonality metrics. The authors define a set of seasonal metrics based on sea ice concentration and surface temperature simulated by CMIP6 and CESM earth system models. A recent satellite database describing several ice seasonality metrics is also used to evaluate models.

A first part is focused on transition period between the different melt and freezing dates. They find an asymmetry between spring ice loss and fall ice growth in both satellite observations and model simulations. They also show that most models have a late freeze onset compared to observations.

In the second part the correlation between seasonal metrics and sea ice area and thickness is studied. The authors find a good correlation between freezing dates , sea ice area and thickness. These correlations allow to highlight sea ice biases in some models which are compensated by other processes. The authors give the example of CNRM model which has biases in melting and in sea ice thickness but which represent realistic sea ice area for the wrong reasons.

I found this paper very interesting. It includes newly observation database and suggests an interesting and novel approach to evaluate earth system models sea ice simulations. I stress the good work the authors made by analysing a large set of earth system models with several members besides satellite observations. I think that this work can make a great contribution to the literature after following comments are addressed in the context of a minor revision.

> We thank the referee very much for their constructive comments. We have made all recommended changes, including adding the suggested new Figures and Table.

*Specific Comments*

1. P1 line 13 "the spread between climate model projections of sea ice has been on the order of millions of square kilometers in Coupled Model Intercomparison Project (CMIP)" . Can you specify about which parameter you are talking? Sea ice coverage ?

> As suggested, we have changed the sentence to clarify that we are talking about sea ice area here. The new sentence reads: "the spread between climate model projections of sea ice area has been on the order of millions of square kilometers in Coupled Model Intercomparison Project (CMIP)".

2. P4 line 92 "(select figures using all available members are provided in the Supplement)." Can you specify which figures ?

> As requested, we have added which figures this refers to: "(a version of Figure 8 using all available ensemble members is provided as Figure S2 in the Supplement)."

3. P9 to P17 : I think it would be beneficial to add maps of the four intra-seasonal periods (melt , freeze , seasonal loss-of-ice and seasonal gain-of-ice periods) and a table of spatial median ( and standard deviation) for each model and observation as for table 4. Moreover, you look at the difference between the spatial of the

metrics to describe the median of the intra-seasonal periods. But as median(A) - median(B) /= med(A-B), calculating the intra-seasonal period for each pixel before doing the spatial median seems more appropriate.

> As suggested, a table of the mean values and model spreads of the intra-seasonal periods has been added as Table 4 in the manuscript. Spatial plots for both intra-seasonal periods and inter-seasonal periods have been provided in the Supplementary (Figues S4-S10). Modeled standard deviations generally agree with satellite data standard deviations. This has been noted in text and the standard deviations have been provided in Supplementary Tables S3 and S4.

> With respect to the differences, we have added text to the manuscript to clarify this process: "As with the transition dates, the inter-seasonal and intra-seasonal periods are calculated at each grid cell before taking the area-weighted spatial means."

4. P11 lines 205-207 "the model spread (May 15- June 3) " : Is it really June 3 or is it June 13 here?

> Excluding the CNRM models, the latest satellite-era median melt onset date occurs on June 3 (ACCESS-CM2 and CanESM5). We have edited the text so that this is clearer.

5. P14 line 228 : What do you mean by "internal variability of the satellite data" ?

> We have rearranged this sentence to say "The maximum range in mean freeze onset dates due to internal variability is 11 days (Table 3) and the majority of the model means (ten out of sixteen) are more than 11 days later than the satellite data, indicating that this delay of the mean freeze onset in the models is not only due to internal variability." The original sentence with the uncorrected values was, "Only five of the sixteen models fall within the maximum range of internal variability (10 days) of the satellite data (Table 3)."

6. p20 line 302 " (Supplementary Table S3)" : I guess you mean Table S4.

> The supplementary table numbers have changed due to the addition of new tables, and in Specific Comment #13 below, we address how we have added text to clarify differences between the tables to avoid confusion. In the original submission, this did refer to Table S3.

7. p22 line 320 "This lack of relationship is a strong indication that the spatial coverage of break-up dates is not sufficient for describing pan-Arctic sea ice feedbacks. " : I wonder if the lack of relationship between March mean ice thickness and break up date can be explained by the inverse relation between ice growth and thickness which explains that the thinner the ice, the more efficient the growth. This relation should temper the delay in break up (see Bitz & Roe, 2004 and Lebrun et al. 2019 ).

Bitz, C. M., Holland, M. M., Hunke, E. C. and Moritz, R. E.: Maintenance of the Sea-IceEdge, J. Climate, 18(15), 2903?2921, doi:10.1175/JCLI3428.1, 2005.

Lebrun, M., Vancoppenolle, M., Madec, G. and Massonnet, F.: Arctic sea-ice-free season projected to extend into autumn, The Cryosphere, 13(1), 79-96, doi:https://doi.org/10.5194/tc-13-79-2019, 2019.

As suggested, we have evaluated the relationship between March mean ice thickness and break-up in the context of the listed papers, and we have added text related to the break-up dates, as well as text related to the freeze-up dates, where this also applies:

LINE 345: "Increases in ice thickness after March may dampen the relationship between thin March ice and an earlier break-up date, since some models show faster ice growth from March to April in areas of thin March ice rather than thick March ice (supporting past work on ice growth rates (Bitz and Roe, 2005)). However, this pattern is not seen in all models and thus cannot fully account for the weakness of the relationships between March ice thickness and break-up."

LINE 330: "With respect to the other fall transition metrics, we find that statistically significant correlations between March ice thickness and freeze-up/closing (which are both based on ice concentration) are less consistent between models, and generally stronger for the closing dates rather than freeze-up dates (Table 7). Other relationships involving freeze-up and spring sea ice of the following year, such as the relationship between the timing of freeze-up and the next year's break-up, have been shown to be dampened by the tendency of thin ice to grow faster than thicker ice (Bitz and Roe, 2005, Lebrun et al., 2019). The growth rate of thin ice, in addition to the spatial coverage of the freeze-up dates, may be limiting the impact that a late freeze-up date has in reducing the following year's March mean ice thickness."

8. p22 line 247 " indicating that the impact of seasonal transition biases can be be large" you should remove a "be"

Corrected.

9. Figure 2 : This figure seems not describe in detail in the main text. You should move it in supplementary.

As suggested, we have moved Figure 2 to the Supplement (now Figure S3).

10. Figures 3 and 6: Can you remind the definition criteria for melt and freeze onset dates in both caption as you did for fig 4,5,7 and 8 ?

As suggested, we have added the phrase "(defined using surface temperature in the models and brightness temperatures in the satellite data)" to the captions of these figures (now Figures 2 and 5) to clarify how the melt and freeze onset dates are derived.

11. Table 3 : What do you mean by "spread" ?

Here the word "spread" is referring to the difference between the earliest and the latest dates found between the first member of all models (the all-model spread) and using the first thirty members of each model (the models marked with * ). To clarify this, we have now defined "spread" in the Methods section and in the captions of Tables 3-5 that refer to spread.

12. Table 3 - Table 4 : Can you also add a spatial standard deviation for each model and observation?

As discussed above, modeled standard deviations generally agree with satellite data standard deviations. This has been noted in text and the standard deviations have been provided in Supplementary Tables S3 and S4.

13. Table 6 or Table S4 : Caption for both tables are exactly the same. I guess it is a mistake you should fix.

We have added text to clarify what is different for figures and tables with very similar captions. For example in Table S6 (table numbers have changed due to the addition of new tables) we have added, "As in 
[revised manuscript text omitted]

**Figure S4.** The length of the melt period (number of days between melt onset and opening) averaged over 1979–2014 at each grid cell using satellite data (a), the first ensemble member of the CESM LE (b) and the first ensemble member of each CMIP6 model (c–q). Stippling indicates where closing dates exist in less than 20% of years in the time range. Models on tripolar grids produce plot gaps filled by gray lines. Negative values indicate where the opening date falls earlier than the melt onset. This can occur due to physical reasons (i.e., dynamical ice divergence or bottom melt), or due to the fact that melt onset is defined using surface temperature and opening is defined using ice concentration.

[Figure]

**Figure S5.** The length of the seasonal loss-of-ice period (number of days between opening and break-up) averaged over 1979–2014 at each grid cell using satellite data (a), the first ensemble member of the CESM LE (b) and the first ensemble member of each CMIP6 model (c-q). Stippling indicates where closing dates exist in less than 20% of years in the time range. Models on tripolar grids produce plot gaps filled by gray lines. No negative values are possible as both metrics are based on sequential ice concentration thresholds (80% and 15%).

[Figure]

**Figure S6.** The length of the freeze period (number of days between freeze onset and freeze-up) averaged over 1979–2014 at each grid cell using satellite data (a), the first ensemble member of the CESM LE (b) and the first ensemble member of each CMIP6 model (c-q). Stippling indicates where closing dates exist in less than 20% of years in the time range. Models on tripolar grids produce plot gaps filled by gray lines. Negative values indicate where the freeze-up date falls earlier than the freeze onset. This can occur due to physical reasons (i.e., dynamical ice convergence), or due to the fact that freeze onset is defined using surface temperature and freeze-up is defined using ice concentration.

[revised manuscript text omitted]